# Machine Learning Study in Caries Markers in Oral Microbiota from Monozygotic Twin Children

**DOI:** 10.3390/diagnostics11050835

**Published:** 2021-05-06

**Authors:** Esther Alia-García, Manuel Ponce-Alonso, Claudia Saralegui, Ana Halperin, Marta Paz Cortés, María Rosario Baquero, David Parra-Pecharromán, Javier Galeano, Rosa del Campo

**Affiliations:** 1Facultad de Ciencias de la Salud, Universidad Alfonso X El Sabio, Villanueva de la Cañada, 28691 Madrid, Spain; estheraliagarcia@yahoo.es (E.A.-G.); mpazcor@uax.es (M.P.C.); mbaquart@uax.es (M.R.B.); dparrpec@uax.es (D.P.-P.); rosacampo@yahoo.com (R.d.C.); 2Servicio de Microbiología, Hospital Universitario Ramón y Cajal and Instituto Ramón y Cajal de Investigaciones Sanitarias (IRYCIS), 28034 Madrid, Spain; lugonauta@gmail.com (M.P.-A.); claudiasaralegui1994@gmail.com (C.S.); ana.halperin@gmail.com (A.H.); 3Departamento de Biología, Servicio de Criminalística, Dirección General de la Guardia Civil, 28003 Madrid, Spain; 4Complex Systems Group, Universidad Politécnica de Madrid, 28040 Madrid, Spain

**Keywords:** machine learning, oral microbiota, LEfSe, PCoA, alloprevotella, prevotella, core microbiota

## Abstract

In recent years, the etiology of caries has evolved from a simplistic infectious perspective based on *Streptococcus mutans* and/or *Lactobacillus* activity, to a multifactorial disease involving a complex oral microbiota, the human genetic background and the environment. The aim of this work was to identify bacterial markers associated with early caries using massive 16S rDNA. To minimize the other factors, the composition of the oral microbiota of twins in which only one of them had caries was compared with their healthy sibling. Twenty-one monozygotic twin pairs without a previous diagnosis of caries were recruited in the context of their orthodontic treatment and divided into two categories: (1) caries group in which only one of the twins had caries; and (2) control group in which neither of the twins had caries. Each participant contributed a single oral lavage sample in which the bacterial composition was determined by 16S rDNA amplification and further high-throughput sequencing. Data analysis included statistical comparison of alpha and beta diversity, as well as differential taxa abundance between groups. Our results show that twins of the control group have a closer bacterial composition than those from the caries group. However, statistical differences were not detected and we were unable to find any particular bacterial marker by 16S rDNA high-throughput sequencing that could be useful for prevention strategies. Although these results should be validated in a larger population, including children from other places or ethnicities, we conclude that the occurrence of caries is not related to the increase of any particular bacterial population.

## 1. Introduction

The microbial colonization of the oral cavity starts immediately after birth, differencing among early colonizers (*Streptococcus*, *Veillonella* and *Lactobacillus*), constant (*Gemella*, *Granulicatella*, *Haemophilus* and *Rothia*) and late colonizers (*Actinomyces*, *Porphyromonas*, *Abiotrophia* and *Neisseria*) [1,2,3]. The establishment of this ecosystem and its further composition is influenced by numerous factors as the mode of delivery, diet and antibiotic consumption [4]. Oral health is not only a local stomatological problem, but also an important driver of systemic health, as it has been linked to numerous disorders of the digestive, cardiovascular and genitourinary tracts [5,6,7].

Caries is the most prevalent human disease worldwide, although its incidence varies according to geography and ethnicity [8], and it has conventionally been attributed to the direct action of acidogenic bacteria such as *Streptococcus mutans, Lactobacillus* and *Bifidobacterium* since these microorganisms have been isolated from the lesions. The application of molecular tools based on high-throughput sequencing of the 16S rDNA gene has revealed that microbiota associated to caries is a much more complex ecosystem than expected (http://www.homd.org/, accessed on 5 May 2021) [9]. While traditional studies classified bacteria as pathogens or commensals according to their potential etiological role on diseases, greater focus has been put on the new concepts of eubiosis/dysbiosis and the disbalance of alkalinogenic/acidogenic bacteria in the caries [3,4]. In addition, metatranscriptomic analyses have permitted to extend the cause of oral diseases as periodontitis from the action of a single microorganism to the metabolic activity of the entire ecosystem [10]. Consequently, the number of microorganisms linked to caries has increased considerably in the last decade [11,12,13], including *Streptococcus, Lactobacillus, Veillonella, Actinomyces, Granulicatella, Leptotrichia, Megasphaera, Olsenella, Shuttleworthia* and, most recently, *Scardovia, Atopobium* and *Selemonas* [14]. One of the major challenges is to identify early markers of caries in order to monitor and prevent this disease during childhood. The exploration of biomarkers in saliva has already demonstrated its usefulness in other pathologies [7,15].

Due to all this complexity in the detection of caries markers, we think that a predictive analysis using machine learning tools can be a good starting point in the study of caries using the oral microbiota. Despite the fact that the implementation of Artificial Intelligence (AI) is still far from being completely common in oral health, some studies highlight the improvements that its use would imply in different areas [16].

The rationale of the present work was to identify bacterial biomarkers in saliva for early caries detection. For this purpose, we explored by massive 16S rDNA sequencing combined with robust bioinformatics tools, statistical analysis and machine learning the oral microbiota of monozygotic twins with and without caries.

## 2. Materials and Methods

### 2.1. Patients and Samples

Twenty-one pairs of monozygotic twins were recruited by the first author EAG and divided into two categories: (1) caries group where only one of the twins had caries (22 infants, 73% females, median age of 9 years, range from 6 to 12 years); and (2) control group where neither of them had caries (20 infants, 70% females, median age of 6.7 years, range from 4 to 12 years) (Table 1). Infants were enrolled in 2018 from January to May in four different dental clinics of Madrid (Spain) within the context of their orthodontic treatment. Each child contributed with a single oral lavage sample after 5 min of vigorous rising with 10 mL of sterile water. Samples were immediately frozen after collection and stored at −80 ${}^{\circ }$C until processing. The inclusion criteria were twins aged 4–12 years whose parents and they accepted to participate in the study. In the caries group kids with clear lesions as well as pre-cavity lesions, mainly white spots, were included, whereas other types of lesions were excluded. All participants were adequately instructed to avoid teeth brushing, food and sugar drinks intake during the 2 h before sampling.

### 2.2. Oral Microbiota Characterization

Oral lavages were slowly defrosted at −20 ${}^{\circ }$C during 24 h, followed by another 24 h at 4 ${}^{\circ }$C, and centrifuged at 14,000 r.p.m. for 15 min discharging the supernatant. Total DNA was obtained from the pellet with the Speedtools tissue DNA extraction kit (Biotools), determining their concentration and quality by Qubit fluorometer (Thermo Fisher Scientific, MA, USA). DNA samples were sent to FISABIO (Valencia, Spain) for massive sequencing (2 × 300 bp, MiSeq, Illumina. Cod. 15044223 Rev. A) of the V3 and V4 regions of the 16S rRNA gene, which were amplified with the following primers: Forward Primer: 5${}^{\prime }$-TCGTCGGCAGCGTCAGATGTGTATAAGAGACAGC CTACGGGNGGCWGCAG; and Reverse Primer: 5${}^{\prime }$-GTCTCGTGGGCTCGGAGATGTGTATAAGAGACAGGACTACHVGGGT ATCTAATCC. Sequence quality was measured according to the following parameters: minimum length, 250 bp; trimming quality measure type, mean; trimming quality number from 3’ extreme, 30; and trimming quality window, 10 bp. Shannon–Weaver and Chao1 indexes were used for bacterial alpha diversity estimation excluding taxa with three or fewer reads. Taxonomic affiliations were assigned using the Silva 119 database, and reads with an RDP score below 0.8 were assigned to the upper taxonomic rank, leaving the last rank as unidentified. Relative abundance and contingency tables of the operational taxonomic units (OTUs) included singletons and very low-represented taxa.

### 2.3. Statistical Analysis and Machine Learning Modeling

Statistical analysis was performed using R statistical software v3.5.3. Quantitative data of the reads were homogenized using their relative percentage from the total reads of each sample to allow the comparison between samples. Finally, the Galaxy Huttenhower Platform (http://huttenhower.sph.harvard.edu/galaxy, accessed on 5 May 2021) was used to calculate the Linear Discriminant Effect Size Analysis (LEfSe) algorithm to identify which microbial taxa explain significant differences among groups of samples [17]. The PCoA analyses were performed by Past 3.0 software. Raw sequences were deposited in the GenBank database as Bioproject PRJNA643173.

Simultaneously, we carried out a statistical exploratory analysis to later search for a machine learning model for a possible caries prediction. To carry out this analysis, we ruled out bacterial species with fewer than 50 data with non-zero values. Exploratory analysis was performed using own software in Python. Machine Learning models were developed with Orange3 v3.27 [18]. We carried out different classification models in two ways. In the first case, we used healthy, control and cavity sample labels. In the second case, we only used healthy and caries labels to classify our samples. Using k-cross validation (k = 10), we tested five different classification model: Random Forest, Neural Network, Support Vector Machine, KNN model and a logistic regression.

To evaluate the results of the used algorithms, we used:Classification accuracy is the proportion of correctly classified examples.F-1 is a weighted harmonic mean of precision and recall.Precision is the proportion of true positives among instances classified as positive, e.g., the proportion of cavity correctly identified as cavity.Recall is the proportion of true positives among all positive instances in the data, e.g., the number of cavity among all diagnosed as cavity.

## 3. Results

Both groups of participants were comparable in demographic and anthropometric terms, and all were recruited during their orthodontic treatment without previous suspicion of caries. Oral lavages were processed in a single session and the 16S rDNA massive sequencing was developed successfully, passing the quality filters with adequate negative controls. The numbers of read counts were comparable for all samples. The alpha diversity was analyzed by the Shannon–Weaver and Chao 1 alpha diversity indexes showed no significant differences between groups, but more disperse values were detected in the caries group (Figure 1).

Phyla distributions showed a preserved pattern for each pair of twins, including those from the caries group (Figure 2). Children with caries had a similar phyla distribution to their healthy siblings, whereas controls presented higher proportions of Firmicutes and lower proportions of Proteobacteria.

Up to 119 genera were identified, although 13 of them accounted for 90% of the total abundance [*Streptococcus* (≈30%), *Prevotella* (≈10 %), *Neisseria* (≈9%), *Veillonella* (≈8%), *Gemella* (≈7%), *Haemophilus* (≈6%), *Alloprevotella* (≈5%), *Rothia* (≈5%), *Porphyromonas* (≈2%), *Fusobacterium* (≈2%), *Leptotrichia* (≈2%), *Granucalicatella* (≈2%) and *Actinomyces* (≈2%)]. The remaining genera represented 10% of the abundance, comprising 106 genera with a total population density less than 1 for each one (Figure 3).

To obtain a global overview of the oral microbiome complexity, we designed an interaction network representing all taxa detected for each subject in circles proportional to their frequency and joined the circles, called nodes, proportional to their frequency joining the circles by lines, called links, to build a network per sample. Subsequently, we superimposed all the individual networks to define the core of the microbiome of each condition (caries, health and controls), and the thickness of the links between nodes is the accumulated number of lectures in all samples, representing the stability in the coexistence of the connecting taxa (Figure 4).

PCoA analysis separated the healthy, caries and control groups (Figure 5), showing a higher level of *Veillonella, Prevotella* and *Fusobacterium* genera linked to a healthy status, whereas *Alloprevotella* and *Granullicatella* were the most differentiated genera among children with cavities. The control group was allocated in a separate quadrant marked by the abundance of *Capnocytophaga, Lautropia* and *Streptobacillus*. Curiously, most of the control group twins were located on the same quadrant (8 out of 10 pairs), three pairs being located in Quadrant 3 (dominated by *Gemella* and *Haemophilus*), three in Quadrant 4 (*Streptococcus* and *Rothia*) and two in Quadrant 1 (*Prevotella* and *Veillonella*). The remaining two pairs of control twins were located on separated coordinates (Quadrants 1–3 and 2–3). Considering the twins of the caries group, only 6 out of the 11 pairs had both children located in the same quadrant: one in Quadrant 1 (*Prevotella* and *Veillonella*), three in Quadrant 2 (*Neisseria, Alloprevotela* and *Leptotrichia*), one in Quadrant 3 (*Gemella* and *Haemophilus*) and one in Quadrant 4 (*Streptococcus* and *Rothia*). The remaining five pairs were distributed in separated quadrants: two pairs in Quadrants 1–2, one pair in Quadrants 2–3, one in Quadrants 1–4 and one in Quadrants 1–3.

Furthermore, differential abundance analysis on microbiota composition by LEfSE in relation to the group, age and the sex of children did not obtain any significant result.

Finally, to address the possibility to predict the cavity in patients, we developed five classification models using machine learning tools (Table 2). The model that showed the highest classification accuracy (CA) was the Random Forest model with a value of 0.881 followed by the Neural Network with 0.810. Studying the confusion matrix, from the point of view of caries, the Random Forest model does not produce false positives but does generate quite a few false negatives (54.5%). On the contrary, Neural Network model produce 16.1% false positives but a lower percentage, with respect to the Random Forest, of false negatives (27.3%).

## 4. Discussion

In the last years, the etiology of caries has evolved from a simplistic infectious perspective (*S. mutans* and/or *Lactobacillus* colonization) to a multifactorial disease involving oral microbiota, human genetic background and environment [3]. In accordance, research tools have evolved from culturomics to metagenomics, transcriptomics and proteomics. Diet continues to be one of the most decisive factors in caries incidence and accounts for the individual susceptibility in relation to carbohydrates intake and bacterial fermentation [19,20], whereas human genetic background seems to be not so relevant, as previously expected [4,21]. The main objective of our work was twofold, on the one hand, to detect early metagenomic markers based on the abundance of particular genera in the oral microbiota associated with caries in the context of monozygotic twins with the same genetic, dietary and environmental context, and, on the other hand, address a classification model to predict caries using this microbiota of patients. When the participation in the project was offered to children, none of them had been previously diagnosed of caries. The bioinformatic analyses showed a more uniform microbiota in the control twin group, but without statistical significance. We were unable to identify any bacterial taxon exclusive of participants with caries, discarding the contribution of oral bacteria microbiota at the initial cariogenic process.

To obtain a representative sample of the entire oral microbiota, all children refrained from tooth brush and avoided food intake for at least 2 h. Despite the high and continuous contamination of the oral microbiota with foreign environmental microorganisms, intra-individual particularities of saliva microbiota have been postulated as a forensic marker to identify subjects, even for twins [22]. Some studies perform the sampling directly from the lesion or at the supragingival plaque, but we decided to use rinsing of the total oral microbiota as a representative sample easily collected by children with the absence of macroscopically visible lesions, which seems to be the most suitable option for surveillance purposes. Even though saliva and supragingival plaque are different in terms of bacterial composition [23], saliva has been used in similar studies, providing differentiation between subjects with and without caries [24]. The use of saliva in the identification of biomarkers associated to both local and general health was previously validated [7,15].

Previous studies on monozygotic and dizygotic twins reported discordant results regarding the incidence of caries and the oral microbiota composition [4,21,23,25,26], although those studies have been conducted in different age groups and using different microbiological methodologies, which could explain the lack of reproducibility. In the last years, tools for massive sequencing data analysis have been evolving considerably, allowing us to applied some of those novel tools to our data, including LEfSE and network analysis of the ecosystem.

Our PCoA analysis consistently associates a higher abundance of *Alloprevotella* in subjects with cavities, whereas in their healthy counterparts *Prevotella* was the most differential marker. Surprisingly, both genera belong to the same family and might have synonymous metabolic functions, although we cannot rule out synergistic effects of combination of microorganisms [27] and, most notably, the interaction of particular bacterial genera with fungi or virus, which has not been extensively explored. Most of the published studies using the entire oral microbiota with a metagenomic approach failed to find significant differences among healthy and caries status in non-related subjects [4,11,24,28,29]. However, a structural conservation between twins can be observed in our PCoA analysis, where control subjects are also more homogeneous (8/10 in the same quadrant) than the caries group (6/11), suggesting an incipient diversification on the oral microbiome.

As in other human ecosystems, the oral microbiome is usually constant and specific in each individual, but may be influenced by ethnicity [30]. A higher prevalence of caries has been described in a group of subjects from China, with special enrichment of *Scardovia* [24], whereas this genus is not particularly abundant in our population. In the study by Yasunaga et al., individuals without caries had more diverse communities, with a significantly higher proportion of the genus *Porphyromonas*, in particular *Porphyromonas pasteri* [31]. Belstrøm et al. also observed a higher alpha diversity in subjects with caries and an enrichment of *Neisseria, Haemophilus* and *Fusobacterium* compared to individuals without caries [29]. In contrast, in our study, both bacterial density and alpha diversity parameters are similar in children of both groups and conditions. An important point is that the composition of the community does not necessarily reflect its metabolic activity [10], particularly in microorganisms represented in low proportions, and it could have essential metabolic activities for the community [5].

Extremely high levels of *S. mutans* have been associated with caries, and, whereas *Streptococcus* was the majoritarian genera, we cannot investigate this point since our metagenomic approach is not able to assign to the level of species, due to the short length of the sequences obtained by massive sequencing. The dominance of *Streptococcus* in the oral cavity can be found in both patients with caries and in controls [29,32], as observed in our case, but being more abundant in the healthy control group than among the cavity group. A protective effect of some streptococcal species has been demonstrated [33], and, beyond the microbiota composition, there is an increasing emphasis on the global ecosystem richness, distribution and functionality [34]. However, in our case, LEfSE analysis failed in the discrimination of children by their cavity status, age, or sex, being the oral microbiota of all participants comparable.

In this study, the classification models showed relatively good precision in predicting caries in our data set (Table 2).The best performing classification models were Random Forest that showed the highest classification accuracy (CA) of a value of 0.881 and closely followed by Neural Networks with an CA of 0.810. One of the most interesting points of the Random Forest model was that it did not produce false positives. However, the worst aspect of these models was the percentage of false negatives (54.5%). On the contrary, the Neural Network model produced 16.1% false positives but a lower percentage, with respect to the Random Forest model, of false negatives (27.3%). This problem could be overcome with a large data-set of caries patients.

The major strength and, at the same time, the major limitation of our study is the inclusive criteria of children, which were enrolled during their orthodontic treatment independently of their caries’ status. In fact, all the detected lesions were small and superficial, corresponding probably to the onset of the disease. Of course, we are unable to ascertain if this was also the case of the control group, where caries were not observed in any children but could be closed to appearing. The oral microbiota could be also implicated in the tooth development [8], and likewise the age of the patient might be considered, since different composition of the oral microbiota has been related to it. Finally, our results show that machine learning models could help us in caries prevention using microbiota data, although they are still far from having good accuracy. In summary, our results demonstrate that composition of oral microbiota in twins is highly conserved independently of their cariogenic process. We were unable to find any bacterial marker by 16S rDNA massive sequencing associated to caries; on the contrary, Isola et al. demonstrated a significant relationship between the salivary IL-6 concentration and existence of periodontitis [7,15].

## Figures and Tables

**Figure 1 diagnostics-11-00835-f001:**
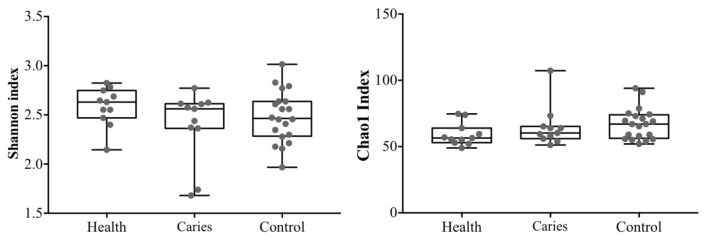
Alpha diversity indexes in all samples. Statistical differences were not detected.

**Figure 2 diagnostics-11-00835-f002:**
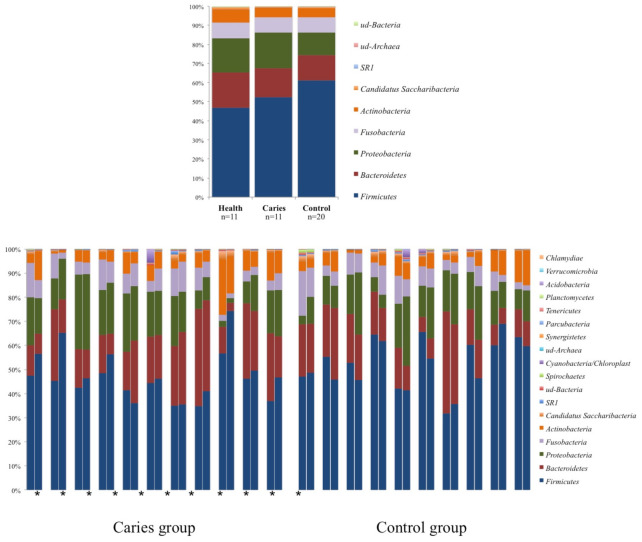
Phyla distribution: (**top**) the median values for each phyla and group; and (**bottom**) all individual values. * Represents Children with caries.

**Figure 3 diagnostics-11-00835-f003:**
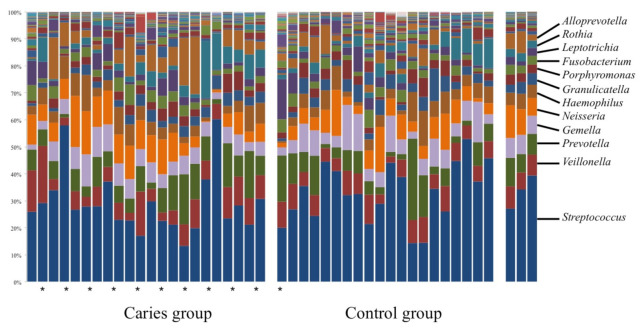
Distribution of the major bacterial genera among all participants.* Represents Children with caries.

**Figure 4 diagnostics-11-00835-f004:**
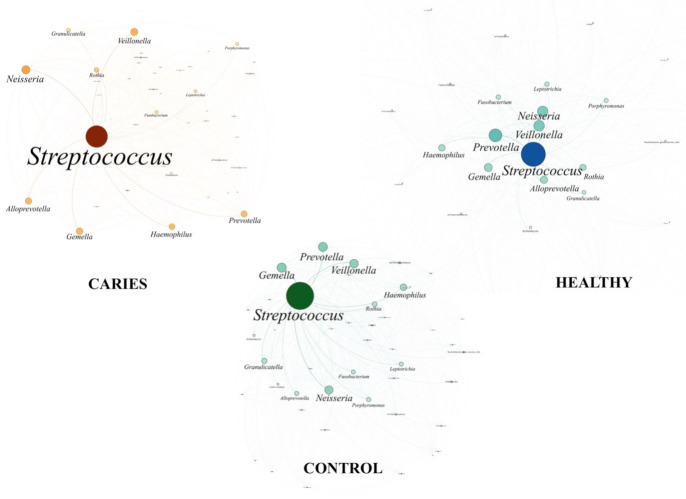
Complex networks core microbiota for the three differenced subjects using Gephi.

**Figure 5 diagnostics-11-00835-f005:**
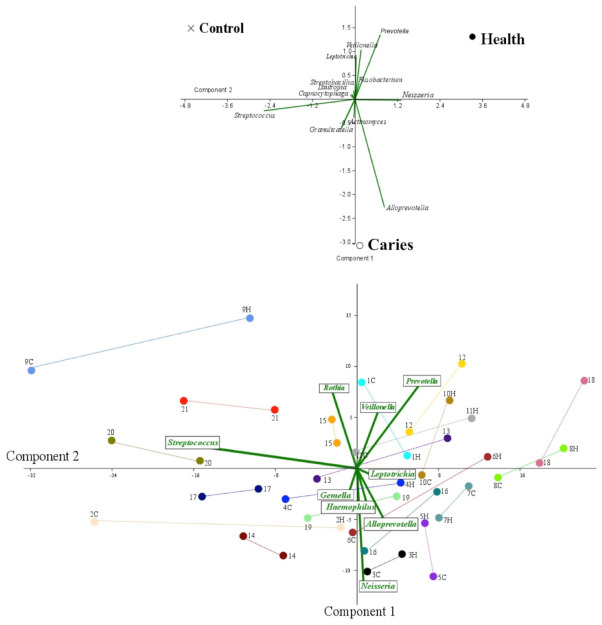
PCoA analysis: (**top**) the median value for the three categories of subjects respect to the abundance bacterial genera; and (**bottom**) the same analysis but considering each of the children and in relation to their sibling. The pairs of twins are linked by colored lines. Random colors are used to highlight the two pairs of twins.

**Table 1 diagnostics-11-00835-t001:** Main characteristics of the 42 participants. C, caries; H, healthy.

Caries Group 22 Infants	Sex	Age
1C/1H	females	12
2C/1H	females	7
3C/1H	females	8
4C/1H	females	6
5C/1H	females	11
6C/1H	males	9
7C/1H	females	8
8C/1H	males	10
9C/1H	females	12
10C/1H	females	9
11C/1H	males	9
**Control Group 20 Infants**	**Sex**	**Age**
12	females	12
13	females	7
14	males	4
15	females	8
16	females	9
17	females	5
18	females	9
19	males	4
20	females	4
21	males	5

**Table 2 diagnostics-11-00835-t002:** Results of the accuracy of the 5 model used in this study. We have used different measures: Classification accuracy (CA), F1, precision, and recall.

Model	CA	F1	Precision	Recall
kNN	0.666	0.655	0.646	0.665
SVM	0.738	0.627	0.545	0.738
Random Forest	0.881	0.874	0.880	0.881
Neural Network	0.810	0.814	0.823	0.810
Logistic Regression	0.595	0.601	0.607	0.595

## Data Availability

Data supporting our study can be found in the GenBank database as Bioproject PRJNA643173.

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
