# Peer review of "Machine Learning Study in Caries Markers in Oral Microbiota from Monozygotic Twin Children"

_diagnostics, 2021, doi:10.3390/diagnostics11050835_

Round 1
Reviewer 1 Report
The manuscript titled “Machine Learning study in Caries Markers in Oral Microbiota from Monozygotic Twin Children” attempted to identify the markers in the oral samples of children with the same genetic, dietary and environmental background, and also to develop a model to predict carries in the microbiota of patients.
As such the manuscript topic is interesting. However, there are some significant deficiencies in the manuscript which need to be addressed to make it publishable:
- Did authors carry out genetic testing to differentiate the twin sets? In other words, is each of the twin sets susceptible to caries in the same way?
- A detail Table of the patient characteristics is needed? Without that it’s very difficult to understand the patients.
- Authors stored the oral samples to -80C. Did the temperature sensitivity affect the viability of the bacterial studied here? Will the results be different if the samples were processed immediately? A validation study is needed to delineate the effect of freezing on viability of bacteria studied in this work.
- Table 1 hasn’t been elaborated in the discussion section.
- Since authors indicate that they were unable to find any bacterial marker that could be useful in future prevention strategies, what is the relevance of this study?
- Page 8, line, 257, “although our results might be validated in a greater number of individuals”. If that’s the case, authors need to expand the study or reanalyze the data to find significant take home message from the current set of data.
Author Response
Question: 1. Did authors carry out genetic testing to differentiate the twin sets? In other words, is each of the twin sets susceptible to caries in the same way?
Answer: The most relevant aspect of this work was to select monozygotic twins in which only one of them has caries. As monozygotic twins, their genetic background is exactly the same, and in addition, the environmental and dietary influences are exactly the same.
Question: 2. A detail Table of the patient characteristics is needed? Without that it’s very difficult to understand the patients.
Answer: The suggested table have been incorporated.
Question: 3. Authors stored the oral samples to -80C. Did the temperature sensitivity affect the viability of the bacterial studied here? Will the results be different if the samples were processed immediately? A validation study is needed to delineate the effect of freezing on viability of bacteria studied in this work.
Answer: Certainly freezing affects the viability of the bacteria, but in this case that factor is not relevant as what we are analysing is the DNA without bacteria cultivation. The technique we use allows us to identify the bacteria correctly regardless of their viability.
Question: 4. Table 1 hasn’t been elaborated in the discussion section.
Answer: We have introduced a new paragraph in the discussion section (lines 266-274) to explain the details of table, now the table 2.
Question: 5. Since authors indicate that they were unable to find any bacterial marker that could be useful in future prevention strategies, what is the relevance of this study?
Answer: Certainly our results are not what we would have initially expected, but we firmly believe that is our obligation to communicate it to the scientific community in order to contribute to the knowledge of caries. Our negative results will help other colleagues in the design of their studies, preventing them from repeating the same hypothesis.
Question: 6. Page 8, line, 257, “although our results might be validated in a greater number of individuals”. If that’s the case, authors need to expand the study or reanalyze the data to find significant take home message from the current set of data.
Answer: The referee is right and we have modified this sentence. Thanks for your comment.
Reviewer 2 Report
In the manuscript entitled: “Machine Learning study in Caries Markers in Oral Microbiota from Monozygotic Twin Children”, the authors evaluated bacterial markers of early caries. To minimise the other factors, the composition of the oral microbiota of twins in which only one of them had caries was compared with their healthy sibling.
The authors found that twins of the control group have a closer bacterial composition than those from the caries group. However, statistical differences were not detected and we were unable to find any particular bacterial marker by 16S rDNA high-throughput sequencing that could be useful for prevention strategies.
The authors concluded that the occurrence of caries is not related to the increase of any particular bacterial population.
Major comments:
In general, the idea and innovation of this study, regards analysis of carie biomarkers of oral microbioma is interesting, because the role of these aspects in dentistry are validated but further studies on this topic could be an innovative issue in this field could be open a creative matter of debate in literature by adding new information. Moreover, there are few reports in the literature that studied this interesting topic with this kind of study design.
The study was well conducted by the authors; However, there are some concerns to revise that are described below.
The introduction section resumes the existing knowledge regarding the important factor linked with biomarkers of oral microbioma.
However, as the importance of the topic, the reviewer strongly recommends, before a further re-evaluation of the manuscript, to update the literature through read, discuss and must cites in the references with great attention all of those recent interesting articles, that helps the authors to better introduce and discuss the role of biomarkers such as suPAR, Galectin 3 and IL-6 during periodontitis : 1) Isola G, Lo Giudice A, Polizzi A, Alibrandi A, Murabito P, Indelicato F. Identification of the different salivary Interleukin-6 profiles in patients with periodontitis: A cross-sectional study. Arch Oral Biol. 2021 Feb;122:104997. doi: 10.1016/j.archoralbio.2020.104997. 2) Isola G, Polizzi A, Alibrandi A, Williams RC, Leonardi R. Independent impact of periodontitis and cardiovascular disease on elevated soluble urokinase-type plasminogen activator receptor (suPAR) levels. J Periodontol. 2020 Oct 22. doi: 10.1002/JPER.20-0242.
The authors should be better specified, at the end of the introduction section, the rational of the study and the aim of the study. In the material and methods section, should better clarify the inclusion and exclusion criteria as well as the initial names of the clinicians involved in the sample preparatrion.
The discussion section appears well organized with the relevant paper that support the conclusions, even if the authors should better discuss the relationship between early mediators of periodontitis regarding the gingival healing post-treatment. The conclusion should reinforce in light of the discussions.
In conclusion, I am sure that the authors are fine clinicians who achieve very nice results with their adopted protocol. However, this study, in my view does not in its current form satisfy a very high scientific requirement for publication in this journal and requests a revision before a futher re-evaluation of the manuscript.
Minor Comments:
Abstract:
- Better formulate the abstract section by better describing the aim of the study
Introduction:
- Please refer to major comments
Discussion
- Please add a specific sentence that clarifies the results obtained in the first part of the discussion
- Page 7 last paragraph: Please reorganize this paragraph that is not clear
Author Response
Question: 1. However, as the importance of the topic, the reviewer strongly recommends, before a further re-evaluation of the manuscript, to update the literature through read, discuss and must cites in the references with great attention all of those recent interesting articles, that helps the authors to better introduce and discuss the role of biomarkers such as suPAR, Galectin 3 and IL-6 during periodontitis : 1) Isola G, Lo Giudice A, Polizzi A, Alibrandi A, Murabito P, Indelicato F. Identification of the different salivary Interleukin-6 profiles in patients with periodontitis: A cross-sectional study. Arch Oral Biol. 2021 Feb;122:104997. doi: 10.1016/j.archoralbio.2020.104997. 2) Isola G, Polizzi A, Alibrandi A, Williams RC, Leonardi R. Independent impact of periodontitis and cardiovascular disease on elevated soluble urokinase-type plasminogen activator receptor (suPAR) levels. J Periodontol. 2020 Oct 22. doi: 10.1002/JPER.20-0242.
Answer: Thanks for your comment, we have incorporated both references and discussed the results.
Question: The authors should be better specified, at the end of the introduction section, the rational of the study and the aim of the study. In the material and methods section, should better clarify the inclusion and exclusion criteria as well as the initial names of the clinicians involved in the sample preparation.
The discussion section appears well organized with the relevant paper that support the conclusions, even if the authors should better discuss the relationship between early mediators of periodontitis regarding the gingival healing post-treatment. The conclusion should reinforce in light of the discussions.
Answer: All suggestions have been added in the new version. Inclusion and Exclusion criteria were already detailed at lines 67-71.
Associated to caries, on the contrary Isola et al. demonstrated a significant relationship between the salivary IL-6 concentration and existence of periodontitis.
Question: In conclusion, I am sure that the authors are fine clinicians who achieve very nice results with their adopted protocol. However, this study, in my view does not in its current form satisfy a very high scientific requirement for publication in this journal and requests a revision before a further re- evaluation of the manuscript.
Answer: We have reviewed the manuscript in detail and hoping that the new version will be adequate, anyway we are open to any suggestion to improve.
MINOR COMMENTS:
Abstract: Better formulate the abstract section by better describing the aim of the study
Introduction: Please refer to major comments
Discussion: Please add a specific sentence that clarifies the results obtained in the first part of the discussion
Page 7 last paragraph: Please reorganize this paragraph that is not clear.
Answer: All minor comments have been incorporated in the new version.
Round 2
Reviewer 1 Report
Thank you for addressing the comments.
Reviewer 2 Report
The authors have well addressed all reviewer's comments.
I suggest the acceptance of this interesting manuscript.